# Analysis of *SNHG14*: A Long Non-Coding RNA Hosting *SNORD116*, Whose Loss Contributes to Prader–Willi Syndrome Etiology

**DOI:** 10.3390/genes14010097

**Published:** 2022-12-29

**Authors:** Shadi Ariyanfar, Deborah J. Good

**Affiliations:** Department of Human Nutrition, Foods, and Exercise, Virginia Tech, Blacksburg, VA 24060, USA

**Keywords:** small nucleolar RNA, UBE3A-ATS, HBIII-85, PET1, PWCR1, cancer, immunity, IL-8

## Abstract

The Small Nucleolar Host Gene 14 (*SNHG14*) is a host gene for small non-coding RNAs, including the *SNORD116* small nucleolar C/D box RNA encoding locus. Large deletions of the *SNHG14* locus, as well as microdeletions of the *SNORD116* locus, lead to the neurodevelopmental genetic disorder Prader–Willi syndrome. This review will focus on the *SNHG14* gene, its expression patterns, its role in human cancer, and the possibility that single nucleotide variants within the locus contribute to human phenotypes in the general population. This review will also include new in silico data analyses of the *SNHG14* locus and new in situ RNA expression patterns of the Snhg14 RNA in mouse midbrain and hindbrain regions.

## 1. Introduction

In the newest release of GRCh38p14 in the e!Ensembl database (July 2022), the number of non-coding genes exceeds the number of protein-coding genes by 5530 (25,134 non-coding genes versus 19,804 protein-coding genes) [1]. The majority of the non-coding (nc) genes identified to date fit within the long non-coding RNA (lncRNA) category (18,049 genes), and, with the exception of a few, we know very little about the roles, regulation, and effects of single nucleotide variants on them. 

LncRNAs are classified as those ncRNAs > 200 nucleotides that are not translated and can exceed 17,000 nucleotides, i.e., *Xist* [2]. Differential splicing and alternative start and stop sites contribute to numerous tissue-specific transcripts for lncRNAs—as many as 170,000 different species of lncRNAs have been annotated [2]. However, the presence of a transcript alone does not need to imply functionality—some of the annotated lncRNA species may simply be the result of enhancer or promoter activity—especially those that have a low expression [3]. Papers on one subclass of lncRNAs, the small nucleolar host gene lncRNAs (SNHG lncRNAs), have increased more than 10-fold in recent years, from 32 in 2016 to 428 in 2021. One of these genes, *SNHG14*, is the subject of this review. In addition to relevant background materials, this review will include new *in silico* data analyses of the *SNHG14* locus, focusing on a cluster of small nucleolar genes it hosts as part of the *SNORD116* locus. The *SNORD116* locus (also called the *SNORD116* cluster, *HBII-85*, *PET1*, and *PWCR1*) encodes 30 small nucleolar C/D box RNAs that are processed from the host gene, *SNHG14* [1]. 

The *SNHG14* locus will be reviewed, and the expression patterns of *Snhg14*/*SNHG14* in whole mouse brain and human RNA seq datasets will be examined. Single nucleotide variants (SNVs), as well as deletions contained within the *SNHG14* locus, and their possible association with human disease in the general population, will be discussed.

## 2. The *SNHG14* Locus

SNHG14 RNA was first identified using reverse transcription PCR experiments, which had been seeking to amplify the *UBE3A* locus [4]. In addition to the expected size using PCR amplification of UBE3A transcripts, the researchers found a larger amplicon of 979 base pairs, representing an abundant, un-spliced form of the UBE3A transcript. The antisense version of this transcript was absent in the RNA from the brains of patients with Prader–Willi syndrome (PWS) but present in the brains of patients with Angelman syndrome (AS, OMIM 105830). The authors hypothesized that, like the XIST long non-coding RNA, the UBE3A-ATS transcript had the ability to compete with the paternal chromosome 15q locus of *UBE3A*, maintaining the expression of the UBE3A mRNA only from the maternal chromosome in tissues, such as the brain. In that same year, a yeast artificial chromosome library was able to complete the gaps in the locus, mapping the rest of the region [5].

In humans, the *SNHG14* locus spans 965,992 base pairs (966 kb) on chromosome 15q11.2, with 148 exons (NCBI NR_14677) and more than 21 different transcripts (Figure 1). The region was mapped, and the snoRNAs were characterized by Cavaille and colleagues more than a decade ago [6]. Multiple alternatively spliced transcripts with alternative promoter usage for the protein-coding genes *SNRPN* (Gene ID 6638) and *SNURF* (Gene ID 8926) exist in a bicistronic transcript originating from different open reading frames on the transcripts. According to the latest updates of these entries in October and November 2022 from both the NCBI and UCSC databases, the entire lncRNA transcript has multiple start and termination sites and undergoes extensive alternative splicing, as described in a 2001 article [7]; yet, the databases state that all of the possible spliced forms from this locus are not currently known nor annotated. An excellent review of the locus from 2017 provides more details on the locus and transcriptions annotated at that time [8].

In addition to the major exons of the locus, more than ten years ago, Yin and colleagues identified a new class of non-coding RNAs that are derived from the *SNHG14* locus, which they called sno-lnc RNAs, as the RNAs were between 1200 and 2900 nucleotides with the SNORD116 snoRNAs at their 5′ and 3′ ends [9] (Figure 1). This class of ncRNAs derived from the *SNHG14* locus was shown to sequester the Rbfox2 splicing factor, allowing for normal developmental processing of the mRNAs [9]. Unlike sno-lnc RNAs, which are not capped or polyadenylated, SPA RNAs were found to overlap this region but are both capped and polyadenylated [10]. SPA RNAs 1 and 2 are 3.4 kb and 1.6 kb, respectively, and act to sequester the RNA binding proteins, such as Rbfox2 and hnRMPM [10]. According to Wu and colleagues, SPA RNAs are made via polycistronic transcription accompanied by RNA Pol cleavage and polyadenylation at a poly A recognition site [10]. Both the SPA2 RNA and the SNO-LNC RNAs overlap part of the *SNORD116* region on the *SNHG14* locus.

The mouse locus for *Snhg14* is on chromosome 7B5, 34.04 cM (Gene ID 52480), and is longer than the human locus at 1,177,441 nucleotides (1177 KB), with 172 exons (Figure 2). Like the human locus, the mouse *Snhg14* locus serves as a host gene for the *Snord116* and *Snord115* sno-RNA loci in mice, as well as the antisense transcript to Ube3A, the protein-coding *Snrpn*, and *Snurf* loci (NCBI NR_146211) (Figure 2), and the MGI database references nine different transcriptional start sites [11] (http://www.informatics.jax.org/marker/MGI:1289201, accessed on 27 December 2022). Interestingly, the mouse locus is missing some of the non-coding RNAs (ncRNAs) found in humans, including SNORD109a and SPA2 lncRNAs. SPA1 lncRNA is retained in approximately the same position as in humans. Copy numbers of the snoRNA clusters also differ [7,12]. More work remains to understand if these differences between the mouse and human locus may translate into differences in the overall function of the locus.

Both the mouse and human loci are imprinted, such that the *SNHG14*/*Snhg14* gene is only expressed from the paternally inherited chromosome. This becomes important for the human conditions that result from the deletion or alteration of the locus. Nearly all known cases of PWS (OMIM 176270) result from the inheritance of a paternal deletion or other genetic situations, such as maternal uniparental disomy, an imprinting defect, or, rarely, chromosome translations. As previously shown, brain tissue from patients with PWS lacks an expression of the *SNORD116* locus by Northern analysis [6], reverse transcription PCR [7], and quantitative RT-PCR [13]. In 60% of the cases, the loss of expression from the *SNORD116* locus is caused by large or small deletions within the locus, with maternal uniparental disomy and epigenetic changes in the DNA’s methylation or loss of the imprinting control region, making up the other 40% of the cases [14]. Regardless of the type, each affects the paternal expression of the *SNORD116* cluster of sno-RNAs (except in one patient with PWS [15]) and, therefore, the SNHG14 transcript expression. Conversely, all the cases of AS lead to loss of the normal *UBE3A* protein expression, either through deletion, uniparental disomy, abnormal imprinting, or missense mutations. 

There are six common breakpoints within the proximal region of the human chromosome 15q [16]. *SNHG14* is located between breakpoints 2 and 3. One PWS patient (patient #5) was recently described and had an atypical 126 kb deletion spanning from *SNRPN* through to *SNORD109a* but excluded the *SNORD116* locus in the deletion (chr15: 25153293-25279455 in hg19) [15,17]. To date, this is the only known patient whose deletion does not include the imprinting control center or the *SNORD116* locus. Chromosome 15q’s rearrangements and breaks are the most common chromosomal changes seen in the human population [18]. Of interest is whether the deletion of any region of *SNHG14* can disrupt the overall mRNA stability and/or splicing of host RNAs. Several studies have addressed this question, finding that gene expression is generally repressed. For example, in a patient carrying a translocation between chromosome 15 and 19 46, XY, t(15;19) (q11.2;q13.3), the lymphocyte expression of the paternally-expressed genes, such as *SNORD116* and *IPW*, were completely absent [19]. 

To summarize, the *SNHG14* locus is large and serves as a host gene for more than 15 coding and non-coding genes. It is complex in producing multiple RNAs from the region, some of which may not yet be annotated. Finally, the deletion of the regions within the *SNHG14* locus, or translocations affecting the expression of the transcripts from the host gene, or uniparental disomy leading to the normal expression of only some genes in the locus, lead to PWS.

## 3. Expression of SNHG14 RNA

Early studies identified DNAse I hypersensitive sites in the proximal region of what was thought to be the start site for *SNRPN* [20]. Further sequencing and analysis of the region have revealed a start site for the SNHG14 transcript, which is slightly upstream of the *SNRPN* locus (Figure 3A), with an annotated promoter and the enhancers enriched in that region. In fact, there are both neuronal and non-neuronal start sites for the SNHG14 transcript, which are described thoroughly and compared with the mouse region in a paper by Chamberlain from nearly a decade ago [21]. In considering the putative promoter region, as shown in Figure 3B, multiple sites for p300 histone acetylase are found here, suggestive of a transcriptionally active region. Indeed, the regulatory build from NCBI identifies multiple putative enhancers and promoters in this region.

Our laboratory recently showed that one of the Snord116 snoRNAs in mouse Snord116-3 could post-transcriptionally regulate the expression levels of Nhlh2, a basic helix-loop-helix transcription factor [22]. We and others have shown that Nhlh2, like other family members, binds to a canonical E-box motif, CANNTG. Chromatin immunoprecipitation of Nhlh2 has identified CAGCTG as a motif used by Nhlh2 in the MC4R promoters [23], and, as shown in Figure 3B, this motif is also found three times in the putative *SNHG14* promoter, with two of the E-box motifs found adjacent to a p300 histone acetyltransferase motif. It is not yet known if the NHLH2 protein can transactivate the *SNHG14* host gene, but if so, then it likely acts in a positive regulatory loop, being increased in the presence of the SNORD116 transcripts and maintaining its own expression by upregulating the host gene transcript, SNHG14. In support of this hypothesis, work from 2004 showed that Nhlh2 expression in the mouse P19 cell line undergoing neuronal differentiation preceded Snhg14 expression (there using the older name Ube3A-ats) by at least one day during in vitro differentiation [24]. However, this study also indicated that Snurf/Snrpn transcripts were detectable in advance of either the Nhlh2 or Ube3A-ats/Snhg14 expression. In the same study, the Snord116 transcripts were detectable at day 5 of differentiation after Nhlh2, which came up by day 3 of differentiation [24]. This pattern is consistent with Snord116 and Nhlh2 transcript expression during mouse development, which can be detected by a Northern blot on day 12.5 (Snord116 [25]) and by in situ hybridization by day 10.5 (Nhlh2 [26]).

**Figure 3 genes-14-00097-f003:**
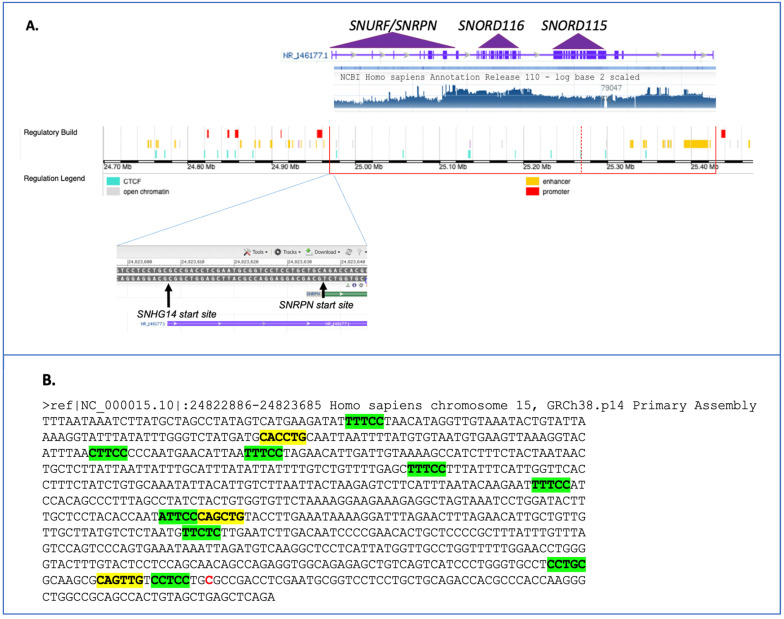
The promoter region of the human *SNHG14* locus. (**A**) Composite figure made from the NBCI genome viewer in the region of the *SNHG14* locus with the RNA-seq and regulatory tracks (top), along with a zoomed-in view of the start sites for the *SNHG14* and *SNRPN* loci. Note the RNA-seq transcriptional activity in regions of the SNORD116 locus and *SNURF*/*SNRPN*. The UBE3A transcript region, nor the other alternative start sites, are in this view. (**B**) The nucleotide sequence surrounding the start of the SNHG14 transcript (shown as a red C residue) was downloaded and examined using the PROMO promoter transcription factor analysis site [27]. Putative p300 sites are highlighted in green. Putative basic helix-loop-helix transcription factor binding sites are highlighted in yellow.

Neuronal tissue generally shows the highest expression for the host gene, *SNHG14* (Figure 4A), but as shown previously by Northern analysis [6], and now in this RNA-seq-generated data [28], the expression is ubiquitous, albeit at different levels. Interestingly, thyroid tissue expression of the SNHG14 transcript is on par (transcripts per million) with that of neuronal tissue. One recent article showed that the SNHG14 host gene RNA expression in thyroid tumor cell lines appears to sponge the microRNA miR-93-5p, resulting in increased tumorigenicity, as measured by cell proliferation, migration, and matrix invasion [29]. The NCBI GEO profiles (GDS3319/1455087_at) from a published study using a mouse model of papillary thyroid tumors indicates a significant increase in the SNHG14 transcript expression in the tumor versus normal tissue [30]. Central hypothyroidism is relatively common in patients with PWS (>10% and up to 30% of patients) [31], and a recent, large retrospective analysis of 110 PWS patients found no increase in thyroid disorders, as measured by the thyroid stimulating hormone level in the population [32], so there is variability. To our knowledge, no patients with PWS have been co-diagnosed with thyroid carcinoma, but this would be expected if SNHG14 expression promotes cancer and is lost in patients with PWS. There was an initial concern that a growth hormone treatment would promote some cancers in patients with PWS, but that does not seem to be the case [33].

SNHG14 RNA is overexpressed in multiple cancer types, including non-small cell lung cancer, hepatocellular carcinoma, glioma, breast cancer, ovarian cancer, and colorectal cancer, among others [34]. These tissues generally have little to no expression of this lncRNA in their normal state. The expression level of this lncRNA correlates with cancer cell proliferation, migration, and the overall aggressiveness of the tumor and, according to a review by Shen and colleagues, with higher levels of the SNHG14 expression promoting the aggressiveness of cancer [34]. While most of the studies in this review are association-based, rather than showing direct causation, several studies in the review demonstrate that miRNAs associated with apoptosis are sponged and blocked by SNHG14 transcripts [34], which may explain why increased expression favors cancerous states. 

The expression levels of the *SNRPN* protein-coding gene, which shares exons and promoters with the *SNHG14* gene, as well as a similar pattern of expression, is very different to that of the SNORD116 expression pattern (compare Figure 4A with Figure 4B,C). For the *SNORD116* locus, this could be due to the need for splicing machinery only associated with NeuN/RbFox3 positive neurons [35]. As the RbFox3 protein is a neuron-specific splicing factor, this suggests that RbFox3 itself is at least in part responsible for the processing of the Snord116 ncRNAs. The GTEX RNA-seq exon-calling method would likely result in the annotation of only the spliced transcripts. For the *SNRPN* gene, a set of alternative promoters that do function in neuronal and other tissues may be responsible for the higher expression of that transcript [24]. Additionally, an excellent set of experiments by Wu and colleagues using a mouse model carrying a 4.8-kb deletion at the PWS imprinting center revealed that the center acts to increase paternal-specific gene expression in the region [36]. 

GTEX analysis also reveals a brain-specific pattern for SNHG14 exon-specific expression, which is not seen in the spinal cord or other tissues (Figure 5). Specifically, three transcripts, ENST00000453082.5, ENST00000554726.1, and ENST00000452731.5, spanning from exon 104 in the locus to the end, have higher expression in the brain than some of the 5′ exons. These are the UBE3A-ATS transcripts. The pattern in the cerebellum is, again, slightly different than that of the other regions (Figure 5). While the GTEX pattern for individual snoRNAs from the *SNORD116* locus was not provided, the transcript levels from the *SNRPN* gene revealed the use of downstream alternative start sites in the majority of the tissues examined, with the upstream exon usage limited to the brain, heart, and arteries (Figure 5).

In our own work, SNHG14 transcript expression is found throughout the adult brain, including the midbrain neurons of the hippocampus, the thalamus, habenula, and hypothalamus (not shown), as well as in the cortical cells, the cerebellum, and the hindbrain (Figure 6). These are all likely neurons, but further co-localization studies need to be conducted to be sure of the neuronal subtypes in each of these areas. RNAScope™ analysis allows for the more fine-tuning of regional expression, co-expression with other genes, and alternative exon usage. This will be an important next step due to the fact that some results suggest that the proper splicing of SNHG14 to form the SNORD116 snoRNAs requires the NeuN protein [35]. 

Close examination of the Snhg14 expression in the dentate gyrus of the hippocampus shows dense labeling of the granular cell layer, with fewer positive cells in the polymorphic and molecular layers (Figure 6B). This is relevant to findings in patients with PWS undergoing MRI, who were shown to have significantly reduced gray matter volume in their hippocampus [37]. The expression pattern and this finding correlate with reports of severe working memory impairment in some patients with PWS [38] and suggest future studies to examine the functional involvement of the SNHG14 transcripts in the hippocampus. 

At increased magnification, the probe location shows what could be nucleolar localization (Figure 6C). Previous work has shown that while Snhg14 transcripts are retained in the nucleus, its host genes, like the processed Snord116 snoRNAs, have a nucleolar expression [39]. For example, Coulson and colleagues also demonstrated that Snord116 locus-derived processed transcripts were localized to the nucleolus using a transgenic mouse carrying a ubiquitously expressed *Snord116* cassette [35]. However, the long non-coding RNA transcript, *Snhg14*, is retained in the nucleus near the site of the transcription [35]. The central function of the nucleolus is in assembling the ribosomes, but more than 30% of the proteins found in the nucleolus are non-ribosomal [40]. This was a curious finding until subsequent work showed that stress responses, such as serum starvation or DNA damage, resulted in the trapping of cellular proteins [41,42]. Whether any of the nucleolar-retained transcripts can interact with these cellular proteins remains to be investigated. In addition, more work is needed using processed transcript-specific probes to determine if the probes we are using are detecting the unprocessed transcript, the processed host gene transcript, or both.

The high expression of the Snhg14 transcript is also present in the lateral habenula (Figure 6D). The habenula’s role in reward processing for food-motivated behaviors is also consistent with the PWS phenotype. The expression is found in individual cells of the hindbrain and is concentrated in the Purkinje cell layer of the cerebellum (Figure 5D–F). Patients with PWS can have strength and movement deficits that are reflected in lower cerebellar white matter volumes detected by MRI [43,44]. As SNHG14 overexpression in cancer can increase cellular proliferation and migration, it is possible that reduced cell bodies (gray matter) or axons (white matter) in some areas of the brain could be directly due to lack of SNHG14 expression in patients with PWS.

In summary, the expression of the SNHG14 lncRNA is ubiquitous, with high expression found in the nervous system. Our work on mouse brain expression analysis for Snhg14 can be used to fine-tune the RNA-seq analysis to specific structures within the brain but should be considered with some caution with regards to what transcript is being detected. First, the probe set used was developed commercially by ACD/Biotechne (Newark, CA, USA) to detect both unprocessed and processed transcripts from the locus, but it may be able to detect some of the processed transcripts, as indicated by the nucleolar-like expression pattern. The expression of Snord116 snoRNA or other ncRNAs from the region will require the development of additional probes for this purpose. The nucleolar-specific co-expression of the Snhg14-processed transcripts with possible RNA targets may also be invaluable in clarifying the functional information of when and where the ncRNAs are working to regulate or modify target RNAs.

## 4. Single Nucleotide Variant Analysis in *SNHG14* and Its Hosted Transcripts

With more non-coding annotated transcripts in e!Ensembl than protein-coding transcripts, genome-wide association studies (GWAS) generally identify many variants within ncRNA transcripts. To be classified as a variant, the change from the reference genome must be found in at least 1 percent of a population. It is estimated that variants occur once every ~1000 nucleotides, with ~4–5 million single nucleotide variants (SNVs) per person [45]. In the e!Ensembl database, 121,571 variants are listed for the *SNHG14* host gene [1]. One way to narrow down the variants that are associated with phenotypes is through a GWAS study. According to the NHGRI-EBI catalog of published genome-wide association studies (the GWAS catalog [46]), there are currently 15 studies examining nine different traits that have identified significant linkage to variants within or near the *SNHG14* locus. As shown in Figure 7, only five of these SNVs actually map within the *SNRPN* locus, and the others are all 5′ of the region. While the upstream SNVs could be linked to the *SNHG14* host gene through a commonly inherited haplotype, for the purpose of this review, we will focus just on those SNVs which are within the *SNHG14* locus. 

Two SNVs are found within the exons of the *SNHG14* region: rs8033133 within the *SNORD116* locus and rs67659323 within the *SNORD115* locus. Using GWAS Central, an association was found for rs8033133 and blood osmolarity with a modest effect size of 0.09 for carriers of the risk allele A (*P =* 9 × 10^6^ in the general population [47,48]. The A allele is found in approximately 34% of the global population. This SNV is intronic, lying between the exons encoding the SNORD116-28 and SNORD116-29 transcripts, not directly affecting the sequence of the snoRNA. There is no known association between PWS and blood osmolarity, but with up to 34% of the general population carrying this allele, further testing of the mechanism and linkage should be carried out. Complicating the analysis of the variants in this region, there would likely be an imprinting effect with only paternally inherited variant carriers showing any associated phenotype.

Rs67659323 is significantly associated (*P* = 3 × 10^9^ ) with immune biomarker levels in bipolar disease. Worldwide, the variant allele G has a general population frequency of 4%, mainly in those of European ancestry. The SNV is found 51 nucleotides downstream of the encoding exons of SNORD115-41 but again intronic for the *SNHG14* locus. Several studies have shown a higher frequency of bipolar disease in individuals with PWS due to maternal uniparental disomy (e.g., Ref. [49]), and this GWAS study may have revealed additional information about the role of the SNORD115-41 transcript and the *SNHG14* host gene in this phenotype for PWS patients, or other carriers of the variant allele in the general population. It would be interesting to determine if carriers of this SNV on the paternal allele are more prone to bipolar psychosis.

The three remaining SNVs in the *SNHG14* gene are intronic-relative to the known hosted genes. However, since GWAS identify areas that are linked to a phenotype and not always the causative allele, it is relevant to examine these further. Rs4906947 is identified as an associated allele for two studies on neuroticism. In a large study focusing on socioeconomic status and educational attainment, the SNV was associated with neuroticism conditioned on educational attainment (4 × 10^8^ and 2 × 10^8^) [50]. In other words, there appears to be a stronger association between neuroticism in those with SNV and lower educational attainment, and the association was found twice in their analysis [50]. A second independent GWAS study of depressive symptoms and neuroticism also found an association with the C allele of this SNV (7 × 10^7^) [51]. The C allele is the major/reference allele and is found in 80% of the population, according to the dbSNP database [45]. This SNV is found between the *SNORD116* and *SNORD115* loci, close to *PWAR1*. Co-morbid psychiatric illness is found in patients with PWS, so this marker may also provide insight into the role of the *SNHG14* host gene and its functions. The same study on educational attainment found a significant association with rs2714720 and neuroticism, but this time in association with self-rated math ability (*p* = 3 × 10^8^) and highest math class taken (*p* = 3 × 10^8^), as well as the Townsend deprivation index (*p* = 3 × 10^8^), a measure of socioeconomic status [50]. This SNV is located just 10,956 downstream of rs4906947 and intronic between *PWAR1* and *SNORD115*. Finally, rs186045664 was found to be associated with myopia in an ophthalmologic GWAS study of over 53,000 individuals of both European and Asian descent (*p* = 5 × 10^6^) [52], using the GWAS catalog [46]. This SNV is also an intron variant located only 1704 bp from the UBE3A coding mRNA. The minor allele G variant is very rare in the general population. In a study using 1000 patients with PWS, there is a significantly higher prevalence of vision problems, but these could also be attributed to hypotonia, a characteristic of the syndrome [53]. However, the point is that GWAS’ results of variants within the *SNHG14* host gene can be of use to understanding both its functional role in general in individuals without PWS, as well as to pointing to specific parts of the transcript that could play a causative role in some of the phenotypes associated with PWS. Finally, and of the direct relevance to the phenotypes of hyperphagia, circadian rhythm imbalance, and sleep disturbances seen in patients with PWS [54], sleep disorders and altered reward processing or addiction tendencies in non-PWS patients could be linked to neuroticism SNVs found in this region. 

To date, 237,982 variants have been identified within the *SNHG14* locus, some of which, described here, are associated with human traits. It is likely that the associated variants are not the causative ones but rather linked to causative variants that have yet to be identified or were not directly tested in the GWAS. Either way, these associations are another way to help to clarify the roles of the lncRNA transcript and its host genes in human phenotypes. 

## 5. Concluding Thoughts

The *SNHG14* gene locus encodes a non-coding RNA, which is a member of the long non-coding RNA host gene family. Multiple small non-coding RNAs, as well as mRNAs, are also encoded or hosted within this locus. This locus is imprinted with only the paternal allele being expressed, providing some insight into its role in imprinted disease. The high expression occurs throughout the brain, but the expression is also found in many tissues of the body. While the function of an active SNHG14 lncRNA is still being characterized, paternal deletion of the locus, particularly those deletions involving transcripts from the *SNORD116 locus*, is associated with the genetic disorder, PWS. A GWAS’ analysis has linked SNVs’ gene locus with bi-polar disorder, blood osmolarity, neuroticism, and educational attainment phenotypes, opening up the possibility that this gene and its non-coding RNA transcript may control many facets of normal brain and body functioning, with the caveat that the paternal inheritance pattern of the locus will make the genetic analysis of any genotype–phenotype relationship more complicated. More basic research into the role of this lncRNA host gene is warranted, with the hopes that we can elucidate its function and how loss through large deletions or individual SNVs leads to neurological and other phenotypes.

## Figures and Tables

**Figure 1 genes-14-00097-f001:**
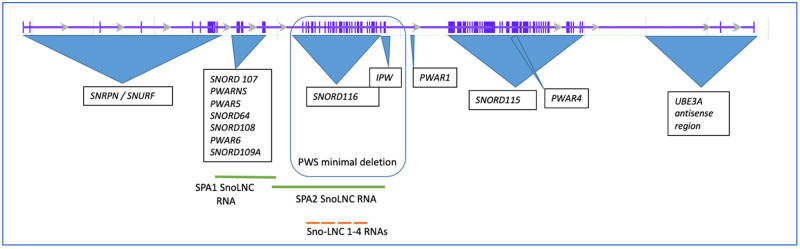
Structure of the human *SNHG14* host gene and associated loci. The initial intron/exon structure of the host gene was obtained from the UCSC browser database, ENST00000656510.1). The minimal deletion that still results in PWS is outlined.

**Figure 2 genes-14-00097-f002:**
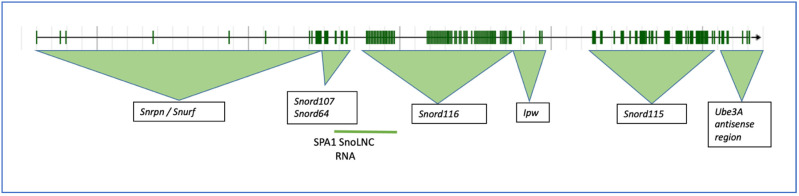
The mouse *Snhg14* host gene on chromosome 7B5 and associated loci. The intron/exon structure of the mouse gene was obtained from MGI, ID 1289201.

**Figure 4 genes-14-00097-f004:**
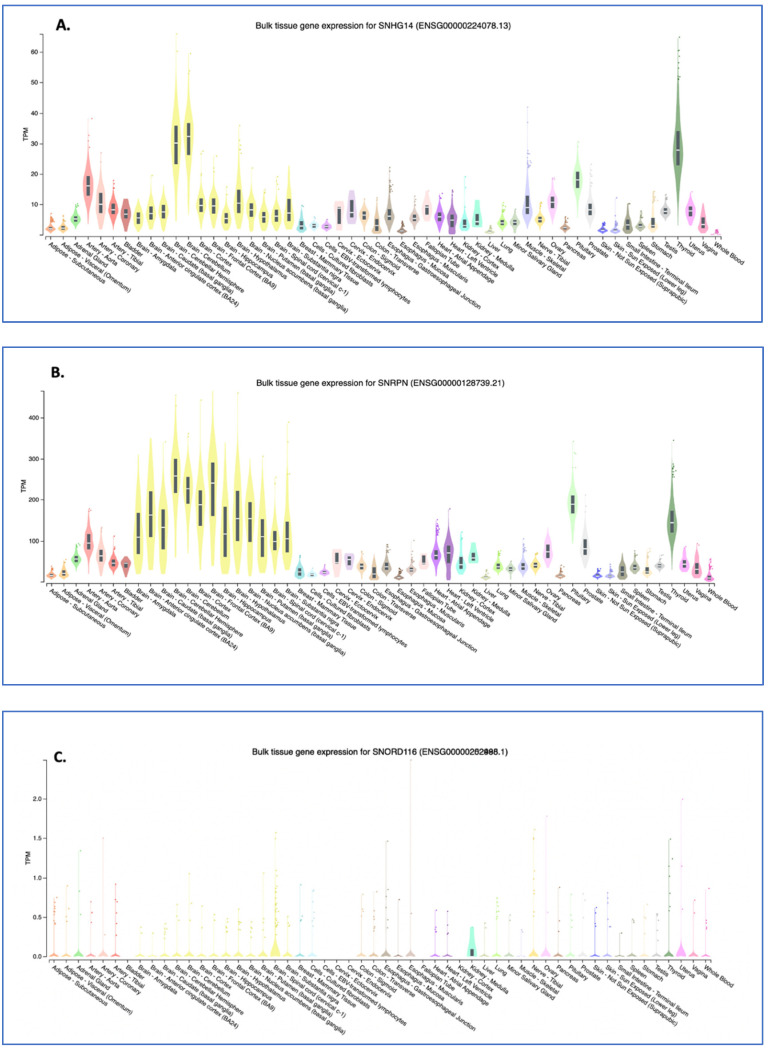
GTEX tissue expression analysis for SNHG14 transcripts. The GTEX expression portal was used to examine the tissue expression pattern for the *SNHG14* host gene (**A**) and two of the genes encoded in the SNHG14 transcript, SNRPN transcripts (**B**), and SNORD116 transcripts (**C**). GTEx Analysis Release V8 (dbGaP Accession phs000424.v8. p2), using 54 tissues and 948 donors for a total of 17,382 samples. For these data, the mRNA from each sample was collapsed to a single number that represents the transcripts for that gene locus per million transcripts in the sample. Thus, individual transcripts are not annotated. The expression patterns are color coded as to tissue type: nervous system-yellow; endocrine-green; skin and muscle-blue; adipose and blood-red; digestive-tan/brown; heart-purple; kidney-turquoise; breast-light blue; female reproductive system-pink; male reproductive system-grey. The tissue types are listed in each section of the figure in alphabetical order. The Genotype-Tissue Expression (GTEx) Project [28] was supported by the Common Fund of the Office of the Director of the National Institutes of Health and by NCI, NHGRI, NHLBI, NIDA, NIMH, and NINDS. The data used for the analyses described in this manuscript were obtained from the GTEx Portal on 09/10/22. TPM, transcripts per million.

**Figure 5 genes-14-00097-f005:**
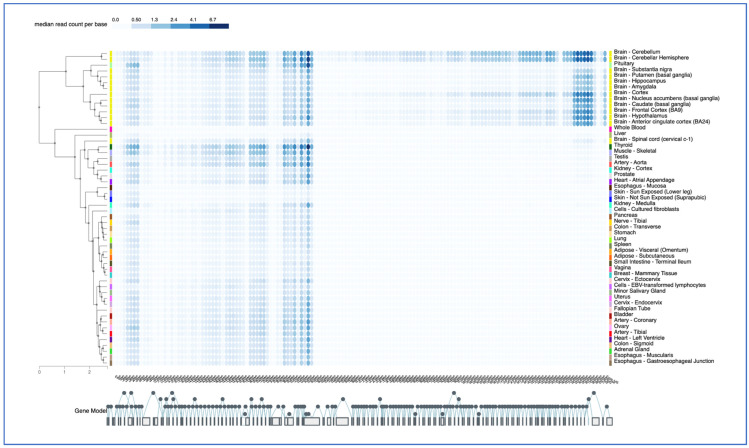
GTEX Exon expression pattern. The GTEX portal was used in the exon expression mode to examine exon-specific transcripts in tissues, color-coded as in Figure 4, and listed in alphabetically order. The 145 exons of the *SNHG14* gene are shown below, with alternative and sequential splicing noted by the filled circles. Alternative splicing is shown as circles above and below the main circles, which are sequential. The Genotype-Tissue Expression (GTEx) Project [28] was supported by the Common Fund of the Office of the Director of the National Institutes of Health and by NCI, NHGRI, NHLBI, NIDA, NIMH, and NINDS. The data used for the analyses described in this manuscript were obtained from the GTEx Portal on 12/28/22. The color code corresponds to read counts per base per exon.

**Figure 6 genes-14-00097-f006:**
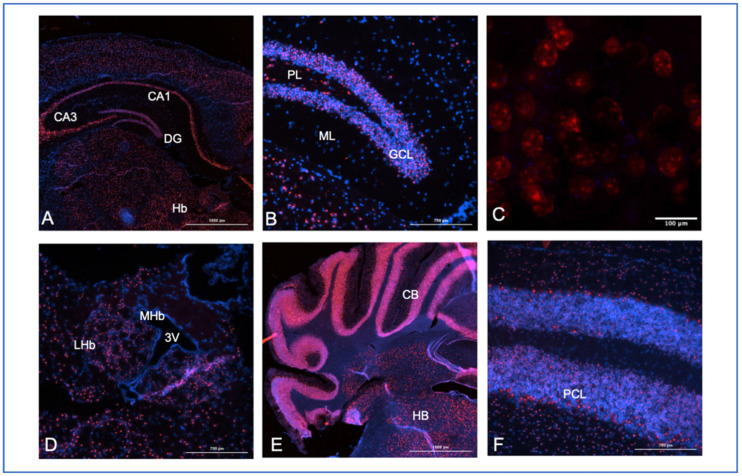
Mouse brain expression analysis for Snhg14. A probe to multiple regions (base pairs 2818–4022 of the NCBI reference sequences, NR_146211.1 and not overlapping any of the known host genes, ACD/Biotechne) for the mouse Snhg14 transcript was used to map expression in brain tissues from perfused animals. Frozen sections (10 mM) from a 28-week-old male mouse were used in the experiments, and all sections were counter-stained with DAPI (blue) with Snhg14 visualized using Opal 620 dyes with a Texas red filter excitation at 560 nm and emission at 610 nm. (**A**) 4× midbrain section showing the CA1, CA3, and dentate gyrus (DG) regions of the hippocampus, along with the habenula (Hb). (**B**) 40× section showing the granular cell layer (GCL), polymorphic layer (PL), and molecular layer (ML) of the dentate gyrus. (**C**) 40× WITH 3× zoom section showing cells in the dentate gyrus. (**D**) Lateral (LHb) and medial (MHb) habenular regions with the third ventricle (3 V). (**E**) Caudal coronal section showing the cerebellum (CB) and hindbrain (HB) regions at 4×. (**F**) 40× section with concentrated expression in the Purkinje cell layer (PCL) of the cerebellum.

**Figure 7 genes-14-00097-f007:**
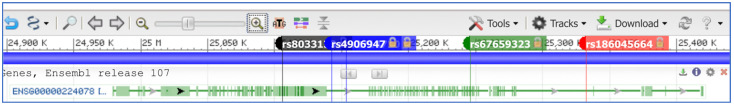
Localization of SNVs within the *SNHG14* host gene locus. The e!Ensembl gene name for *SNHG14* is shown (ENSG00000224078). Five SNVs are shown (rs2714720 is 5′ to rs4906947 with the tag name hidden).

## Data Availability

Not applicable.

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
