# Peer review of "Analysis of SNHG14: A Long Non-Coding RNA Hosting SNORD116, Whose Loss Contributes to Prader–Willi Syndrome Etiology"

_genes, 2022, doi:10.3390/genes14010097_

Round 1

Reviewer 1 Report

Ariyanfar and Good provide a review of expression and genetic variants of the large transcript, SNHG14, the loss of which results in Prader-Willi syndrome (PWS). The authors review current in silico data on RNA expression from the locus and present new expression data in mouse brain. A discussion of potential DNA variants of interest in the gene is also included.

Overall, the review provides some new insight and perspectives on the normal expression and potential function of this important RNA. However, the manuscript would benefit from modifications to more clearly convey some of the points the authors are advancing.  

There is some basic information missing in the introduction. For example ‘SNORD116@’ should be explained -  most papers don’t use the “@” symbol, rather referring to the 'SNORD116 cluster'. Similarly, the manuscript refers to “SNORD116-3” (Line 107) without describing the cluster, nor is it ever stated that the SNORD116 genes encode box C/D small nucleolar RNAs.  The number of repeats in the cluster and basic organization should be explained.  Also, the bicistronic nature of the SNURF/SNRPN transcript should be mentioned.

There is some imprecision in the sentences/language that needs to be corrected. For example, the paragraph including Lines 78-86 is confusing and incorrect as written – PWS doesn’t occur via ‘other alternation of chromosome 15q’.  Beyond paternal deletion, maternal uniparental disomy, imprinting defect or rarely, chromosome translocations, can lead to PWS.  Also, the description that deletion results in “minimally affecting expression of the SNORD116 cluster” makes it seem as if the deletion only minimally impacts expression of the SNORD116 genes, rather than resulting in loss of expression, which is incorrect. It is also not true that all cases of Angelman syndrome lead to loss of UBE3a expression - missense mutations (not mentioned), result in expression of a transcript encoding a mutant protein.  The sentence structure beginning at line 82 results in an incorrect statement - paternal UPD does not result in expression of SNHG14 from the maternal chromosome, it results in SNHG14 expression from both paternal chromosomes (no maternal chromosome present), while maternal deletion results in expression of SNHG14 from the paternal 15 only (the author may not have meant the sentence to suggest that deletion results in SNHG14 expression from both mat and pat chr15, but the sentence currently reads that way) .

Line 88 states that deletions or duplication in this region “have overlapping phenotypes” with AS or PWS, but in fact, deletion in this region causes PWS or AS.  Perhaps, the authors are referring to duplication 15q syndrome (Dup15q)? If that’s the case, it should be clearly stated.

Line 92: No reference is provided when describing the PWS patient whose deletion excludes SNORD116 (presumably ref 5, Grootjen et al).  

Line 135 states there have been ‘isolated’ cases of hypothyroidism in PWS, but the central hypothyroidism is relatively common (reported at ~2-30% across several publications, generally >10%), presumably secondary to hypothalamic dysfunction.

Figure 3 isn’t very clear – the SNRPN and SNORD116@ genes aren’t clearly labeled. The purpose of the expanded graphic of start sites isn’t clear. Is this meant to show different start sites for SNRPN and SNHG14?  The relationship between the transcripts should be more clearly shown and discussed.  Further, use of the multiple start sites should be described – e.g., does the current transcription data support the neuronal vs. non-neuronal transcript differences described almost a decade ago by Chamberlain (PMC3578059)?

The statement that ‘higher levels of SNHG14 promoting aggressiveness of the cancer’ (line 142-143) needs to be supported with an appropriate reference – Has it been demonstrated directly that SNHG14 has a causal role in increasing tumor aggressiveness, or is it an association?   

The detection method for SNORD116 in figure 4b should be described, e.g., does “SNORD116” refer to all of the SNORD116 gene sequences, or a subset of the repeats?

For Figure 5, please describe the elements of SNHG14 that are included in the probe (e.g., Snrpn coding sequences, Snord107, etc).

The significance of the highlighted DNA variants on phenotype of those with PWS (eg, Rs67659323) is difficult to appreciate, given that no individual with PWS (neither PWS due to del nor PWS due to UPD or ICD) would be expected to exhibit expression of the variant allele. This point isn’t very clearly stated. While the variants discussed may influence phenotypes such as psychiatric illness or myopia in the general population, it is difficult to envision how the variants would impact the phenotype of individuals with PWS. Rather, one would anticipate that DNA variants outside the deleted/silenced 15q locus might be more important in influencing the severity of these PWS symptoms. Also, it should be noted that the high degree of vision problems in the PWS population is likely the result of hypotonia that is characteristic of PWS.

Minor comments/edits

Lines 22, 60,  close parenthesis

Line 31 ‘result’ rather than ‘results’

Line 33 – perhaps: “have increased more than 10-fold in recent years, from 32…”

Line 49 – missing a word, “competes with”

Line 70: in reference to SNORD109a and spa lncRNAs, the text states these genes may be in a different region of chromosome 15, but the locus is on chromosome 7 in mouse.

Line 213 – ‘that’ rather than which

Figure 4b has a clipboard icon included.

“Syndrome” in Prader-Willi syndrome is not capitalized. Also, Prader-Willi syndrome should be abbreviated to PWS once, and then referred to as PWS throughout the manuscript.

Throughout the manuscript, beginning on Line 62: “Mouse” should be used rather than ‘murine’, as the focus of the review is mouse only and murine includes mouse and rat.

Long noncoding RNAs are generally referred to lncRNA rather than LNC RNAs, and ‘SPA’ RNAs are capitalized (not spa)

Reviewer 2 Report

 The authors describe the human and murine SNHG14 host gene locus and its gene and lncRNA content in the context of PWS and Angelman Syndrome as well as reviewing the literature on SNPs and disease associations in this region.  New IHC data showing Snhg14 localization are included.  This is a well-written review and analysis of SNHG14.

Three questions would be useful for clarification in the review:

1.       Fig. 5: is this RNA FISH or RNAscope? I think a bit more details is needed on examining Snhg14 localization in mouse brain sections. 

2.       How does UBE3A control SNORD116?  UBE3A seems to be a suppressor, but is the mechanism known?  If not, would you offer possible mechanisms for investigation?

3.       What is the significance of the nucleolar location of SNHG14?  (rRNA comes to mind, but?)

Reviewer 3 Report

The review lacks an overall direction, or message. I don't really understand what the authors are trying to say/add to what is already known. The review is short, and a number of important works are omitted and not discussed at all.

a) Snords clusters discovery and genomic organization of alternative processing of SNHG14: (PMID: 11106375, PMID: 11726556, etc)

b) expression: (PMID: 11106375, PMID: 25246219, etc)

c)localization: (PMID: 20016068, PMID: 35951020 etc)

d)elimination of clusters in mice: (PMID: 18320030, PMID: 18166085, PMID: 33016258, PMID: 35951020)

e) activation of clusters expression: (PMID: 22496793, PMID: 26848093)

etc…

The review covers 3 different aspects (clinical data/patients; expression/tissue-specificity; SNVs) but each of one is rather superficially. None of the three aspects has a clear conclusion.

The main question is what was this review/article supposed to achieve? Was it to publish in situ data?  The authors themselves don't answer that question.

The authors constantly mix different layers of information, e.g. they themselves write "and the SNORD116 ncRNA expression, which is derived from the host gene are very different than the host gene" (lines 148-149) but then talk about features/phenotypes of PWS and other disorder, some of which lack (different parts of) SNHG14 and some of which do not. They also mix up data from mouse and (human) patient, while we know that especially tissue specificity of expression differs between the two species.

The Nhlh2 story, they state that Snord116 increase its mRNA stability (post-transcriptionally) while resent data showed increased expression of Nhlh2 in the Snord116KO brain.

If the authors intend to publish a review, they should present all/most of the available data.

Major points:

P2L51-53: In humans, the SNHG14….     …(Figure 1). The sentence is misleading a) the number of transcripts (RNAs) could be much larger then just 21. b) Alternative splicing is occurring throughout, but not just for SNRPN/SNURF exons c) In humans, SNORD116 and SNORD115 copies differs, forming subgroups (PMID: 11726556), etc…

P3L91-92 …Of interest is whether…

The sentence is not entirely clear, how can deletion of any region of SNHG14 disrupt overall mRNA stability? Which  particular mRNA did the authors have in mind?

 P3 expression of SNHG14 RNA

P3L100-121

This part of the paragraph is confusing as to why the authors showed the human sequence Fig3B, but is mainly talking about mouse. It would have been helpful if the authors had introduced the U-exons.

P5L150-157 For SNORD116, …

The statement in the review is controversial, a) In húmans, SNHG14 expression is ubiquitous and SNORD116 and SNORD115 are efficiently processed in all tissues (PMID: 11106375, PMID: 25246219, etc).

b) In mice, activation of maternal chromosome results in ubiquitous expression of Snhg14 and efficient processing of Snord116 and Snord115 in non-neuronal tissues (PMID: 26848093, PMID: 30862860)

P6L167-188 (including Fig5)

The authors used Snhg14 probe that is antisense to exons hosting Snord107 and Snord64 but not Snord116. Detected transcript may have different regulation (stability etc) than the transcript harboring IpwA exons (exons hosting Snord116 cluster).  

P6L179-180.

At increased magnification, there… nucleolar localization…

Snhg14 is a long non-protein coding transcript that originally shown to localize in nucleus (nucleoplasm) PMID: 20016068. Nucleolar localization is a known fact for snoRNAs that after processing are transported and localized into nucleolus (for ribosomal pre-RNA processing). Although, PWS-locus derived snoRNAs do not have significant complementarity to pre-rRNAs and are most likely not involve in rRNAs modification (PMID: 33016258), they bind to snoRNA-binding proteins and are transported to the nucleolus. Whereas Snord116 and Snord115 host transcript(s) remain in the nucleoplasm (PMID: 20016068).

Consequently, the authors should clarify this issue and explain their observation.

P7. Single nucleotide variant analysis…

The authors should mention that there is no sequence conservation between human and mouse exons (only snoRNAs have some degree of similarities). No clear data are yet available on the function of different Hsa. SNORD116 subgroups.

Although repetitive SNHG14 exons can be clustered based on seq. similarity (IPWA1, A2; G1, G2 etc) there is  are also some degree of variability within clusters. Therefore, the authors should align the exons and see if the two reported SNVs are present in other repetitive exons.

Rs67659323

This SNV located in intron 51nt from the 3’-end of SNORD116-41. Here again, copy sequence analyses of the other snoRNAs may be of interest. However, SNV is less likely to effect SNORD116-41 processing.

P7L237-264

The three remaining SNVs in SNHG14 appear within the transcript but are not directly associated with any of the hosted genes.

It is not clear what the authors had in mind. Are these SNVs intronic?

P9L266-267

I do not agree that the gene locus can be a member of the long non-coding RNA family (Please correct this sentence).

In addition, the PWS locus encodes a number of small nucleolar RNAs (snoRNAs).

 Minor points

P1L34: SHGH14, it should be SNHG14

Figure 1 and Figure 2. The human and mouse loci should be shown in the same orientation from 5’ to 3’.

Figure 1, orientation of transcription is missing.  

P2L64-65. The expressed transcript is on negative strand….

This is a matter of genome annotation, but not transcription. Transcription only occurs in one direction (from 5’-to-3’), so as I suggested above, please correct Fig1 and 2 accordingly.

P2L69-71 …It is not clear if…

Mouse PWS-locus is on chromosome 7 (but not 15).

P9L278...that we can elucidate its normal function,…

I would suggest removing ‘normal’

Round 2

Reviewer 1 Report

The revised manuscript is much improved.  A couple of small edits still needed, including:

Line 39 - delete 'which' (or otherwise modify to make it a complete sentence)

Lines 81-83 - some of this is duplicative with text in the Fig 2 legend, can be deleted

Line 225 - add 'rather' (...rather than showing)

Reviewer 3 Report

P1L9 Small Nucleolar Host Gene 14 (SNHG14) is a host gene for other small non-coding RNAs including SNORD116@ small nucleolar C/D box RNA cluster, SNORD116@

SNHG14 is lncRNAs, please remove “other”.

Why is there an "@" added to the name of the SNORD116 gene? This is confusing. For example, there is no @ in the name of the SNORD115 gene. Please correct throughout the text. Originally, I thought its problems of formatting, but now I realise it’s a problem in GeneCards database (I have contacted them for clarification).

P1L38-39. The SNORD116@ locus (also called the SNORD116 cluster), which encodes 30 small nucleolar C/D box RNAs that are spliced from the host gene SNHG14 (Figure 1).

If I am not mistaken, there are 29 copies of SNORD116 in humans (please check the latest Hsa genome annotation). I would suggest using processed instead of “spliced”. SNORD116 are over 100nts away from the acceptor splice site (AG) and are presumably processed from introns that are already spliced (as suggested by the identification of long ncRNA with snoRNA ends (PMID: 22959273), notably, I did not find discussion of this class of SNHG14 derived transcripts in the review).

P2L55-61 “Others then, showed that when Snord116 is absent (in this case using neuronal cells derived from the paternal Snord116 deletion mice [5]), the Ube3A-ats transcript was increased [6] As a side note, researchers, including our own laboratory who uses the paternal Snord116 deletion mouse model [6] should keep in mind that the Ube3a transcript may also be reduced in these animals. However, this situation should mimic that of humans with 15q deletions involving Snord116”.

This observation from neuronal culture experiments needs more explanation, as in the original publication on Snord116del model Ding, et al 2008 (PMID: 18320030) writes:

Although Ube3a is expressed exclusively from the maternal allele in brain and, therefore, should not be affected by the Snord116 deletion on the paternal chromosome, a reduction of the paternal-specific antisense transcript Ube3aAS could potentially affect silencing of the paternal Ube3a allele and, thus, change its transcript level”.

For another mouse model with Snord116 deletion Rozhdestvensky et al 2016 (PMID: 26848093) reported:

 In line with our previous observation in PWScrp−/m+ mouse models, the expression of IPW-G exons in PWScrp−/m5′LoxP was also reduced (4-fold) in comparison to wild type animals, but the expression levels of Snord115 as detected by RT-qPCR were only slightly lower (1.5-fold)12. This could be due to different stabilities of the RNAs. Notably, the expression level of Snord115 in PWScrp−/m+ mice was almost 2-fold lower than in wild type animals, which was not detected in our previous study using Northern blot analysis12. Interestingly, we also observed a slight decrease of Ube3a antisense transcripts that potentially led to a small increase of Ube3a mRNA isoforms expression in PWScrp−/m+ and PWScrp−/m5′LoxP mouse models.

P3L80-82 The expressed transcript is on the negative strand, as opposed to the human transcript, which is found on the positive strand of the DNA, and so we show in the GenBank orientation (Figure 2); however, strand location only refers to genome annotation and does not affect transcription, which still would occur in the 5’-3’ direction.

Different genome annotations use different stand orientations; a recent annotation of the Hsa genome was made from telomere to telomere (T2T CHM13 assembly DOI: 10.1126/science.abj6987). The authors just need to specify positions from centromere (cen) to telomere (tel) and not confuse readers with sense and antisense transcription, etc.

P3L100-101 As previously, shown, brain tissue from patients with PWS lack expression by Northern analysis [7], reverse transcription PCR [8], and quantitative RT-PCR [10].

This sentence is not complete.

P4L122. .. (hr15: 25153293-25279455 in hg19)

It should be chr15. Also, please use the correct annotation: UCSC Genome Browser on Human (GRCh37/hg19).

and citation:

UCSC Genome Browser: Kent WJ, Sugnet CW, Furey TS, Roskin KM, Pringle TH, Zahler AM, Haussler D. The human genome browser at UCSC. Genome Res. 2002 Jun;12(6):996-1006

P4,5,6… and throughout the text:

It is very difficult to follow the review because the authors mixed gene symbols with RNA and/or protein symbols. I would suggest to stick to the commonly used abbreviation:  

Humans, non-human primates: Gene symbols italicized characters that are all in upper-case (e.g., SNHG14). Protein and RNA symbols are identical to their corresponding gene symbols except that they are not italicized (e.g., SNHG14).

Mice and rats: Gene symbols are italicized, with only the first letter in upper-case (e.g., Snhg14). Protein and RNA symbols are not italicized, and all letters are in upper-case (e.g., Snhg14).

For example P9.L279-312: “In our own work, SNHG14 expression is found throughout the adult brain including the midbrain neurons of the hippocampus, the thalamus, habenula, and hypothalamus (not shown), as well as in cortical cells, the cerebellum and hindbrain (Figure 5)….”

Did the authors investigate the human or the mouse brain? In addition, there are no Material and Methods section and no statement about animal experiments.

P7L246-247 Specifically, three transcripts, ENST00000453082.5, ENST00000554726.1, and ENST00000452731.5 spanning from exon 104 in the locus to the end have higher expression in brain than some of the 5’exons.

all of the above transcripts host for the SNORD115 and SNORD109B genes extending to the UBE3A gene region. In fact, the 3’-exon of ENST00000453082.5 is antisense to the donor splice site of the exon 6 of UBE3A pre-mRNA (I have not checked the others). Hence, these are examples of UBE3A-ATS transcripts.

P9L287-294 “Closer examination of the SNHG14 expression in the dentate gyrus of the hippocampus shows dense labeling of the granular cell layer, with fewer positive cells in the polymorphic and molecular layers (Fig. 5B). This is relevant to findings in patients with PWS undergoing MRI who were shown to have significantly reduced gray matter volume in their hippocampus [29]. The expression pattern and this finding, which is correlated with reports of a severe working memory impairment in some patients with PWS [30], and consistent suggest future studies to examine the with a functional involvement of SNHG14 in the hippocampus…”

As I understood from the author’s response: The probe set used for Snhg14 detected was developed by ACD/Biotechne (Newark, CA) and is proprietary. What the company could tell us is that the probe set is contained between nucleotides 2818-4022, and does not overlap any of the known host genes, and so should recognize the large lncRNA, but not smaller transcripts.”

Does this mean that the authors didn’t know exactly what the probe was detecting? Based on the information in the legend of Fig5, I downloaded the probe’s sequence (see below). Considering this, the probe detected the host transcript for Snord64 and Snord107 but did not detect Snord116-HG or Snord116 host exons. Consequently, as the authors have just mentioned above on P7 describing the differences in the expression of human SNORD115, 109A host transcripts, it could well be the case that the transcript has a detected has different expression pattern (stability) from the Snord116 and Snord115 host transcripts. In fact I can’t recall any previous work using similar probe.The authors should not refer to it as Snhg14, because this is misleading. The authors should clearly define this region of Snhg14 (perhaps a schematic drawing would help)  and better review the manuscript of Vitali et al 2005

atgatagttcatctgtgtggggccttatgaaatttccagtatccgtgtcagtatgtctgctgatggtattgcacagcttctttcatgtctaggagttgctaattaaccctccctcccataatctggaggttggtaaacaggaggctctgatatttggatgtgagtgatgaccacaaatttgaagaaaatctatgatcttattcttgaactaaaagtgaccactcagagttcatgactgaagaaagaaattttcagtgaatgttatgaagtttcatgacttgactaccaccctgtagtattataacaaagctgtaaatcctaatatcgcagcttaagacagaaacataaaccatccaagcaagatgaaatgtccttggatcacaaaaggcctggtgacatggatcctcataaaatggcgttcccaactcatgaaaaaaatggagagccacctttctagatttgtaaggtatatgaatcaggaggaaagggtaatgtttatggagcatgtacattgtggaatgtgtaataattgaagatttgtgcaagtcaattaaactatacatgcttggaaattgccacatgggaccatcttgaatgaacagtttatgcaaacagaaaacatgttcaggaagaatgttaaaattttaacacttgtaaatagaaagaagatgacatttgcctgtctttgtaatttatgagggtatttgcaaataagcatcaaaacaagaagaaaatattaaaaagacattttgtgccaggacatcatggtagtgctggattgcctatccaactctgtataaaggtttttcaaacaatttgtgaagaatgttagtatcttcagagtcttctgcatcagcaaagagtagtgcattaactaatggaaaagagaagataaaatttctggtgctcatgatgactagtccgaacctgtgaacttctgagaaaatttacagttctgaaaactagaggtggttaccgaattgaatagccgttggttctacagccactataatgatctgaaaagttgaatgtgttgcatgaacactttcaggaatcataatagaactcagctaattagatagagactggaatgttgatttgaagacaattactttataaagattttctcattgctgcttttgcattatcccaagaagtggagacactactacatggataagaactgtatgattgaattgactttg

P4L154,P5L155-160. In support of this hypothesis, work from 2004 showed that Nhlh2 expression in the mouse P19 cell line undergoing neuronal differentiation preceded Snhg14 expression (there using the older name Ube3A-ats) by at least one day during in vitro differentiation [19]……

The authors should be more precise about this in vitro correlation; in addition, it is important to mention the regulation of Snords and host transcript(s) expression during mouse embryonic development (expression starts between E12.5-15.5 (for example: PMID: 18166085 and others).

P9L296-298. At increased magnification, there appears to be the transcript shows nucleolar localization (Fig. 5C). This was not surprising as this class of lncRNAs are known to be nucleolar.

The authors should find reports that showing nucleolar localization for “this class of lncRNAs” (rather than rRNAs or those involved in pre-rRNAs or derived from processing or transcribed in antisense to rDNA repeats) (for review PMID: 36207140)

P9L298-299 Previous work has shown Snhg14 and Snord116 nucleolar expression using RNA FISH in rat hypothalamus [31].

This work reported that Snord115 and Snord116 are localised in nucleoli whereas Snord115-HG and Snord116-HG (host transcripts of 116 and 115 or transcripts derived from Snhg14 exons) form clouds in the nucleoplasm (near transcription start sites) - nuclear localisation.

P9L299-301. Coulson and colleagues also demonstrated that SNHG14 was localized to the nucleolus using a transgenic mouse carrying a ubiquitously expressed Snord116 cassette , consistent with other studies [27]. Both articles described retention of Snord116 in nucleoplasm.

The authors should carefully examine Vitali et al (31) and Coulson et al (27) work and do not mislead readers!

Following is a copy-past sentence from Coulson et al (27):

“In addition to the nucleolar snoRNAs, the spliced exons of the Snord116 locus are retained as an RNA cloud that localizes to the site of transcription on the active paternal allele in wild-type neurons (22) (Fig. 1).”

Similarly, to Vitali et. al (31), Coulson et al (27) detected nucleolar signal only for Snord116, whereas Snhg14 exons hosting Snord116 were detected as clouds in nucleoplasm!

P9L312, P10L313-314 These variants make disrupt both the DNA binding activity of NHLH2 as well as the nucleolar localization of the protein and are worth further investigation in both normal and PWS-derived neurons.

Although the authors predicted NoLS in the NHLH2 sequence, no nucleolar localisation of the protein was reported, I would suggest that the sentence should contain an assumption. 

P10L314-315 As we have shown regulation of Nhlh2 transcript stability by Snord116 [17], this possible nucleolar localization of Nhlh2 during stress responses may explain when the regulation occurs.

Did the authors refer to Nhlh2 gene or mRNA? If I am not mistaken it was shown that Snord116 could increase posttranscriptional stability of mRNA. In addition, it is not clear how putative nucleolar localization of Nhlh2 protein could be regulated by Snord116 activity.

P10L317-320 Stress-response based differences in the experimental methodology may also explain why Kummerfeld and colleagues found an increase (rather than a decrease) in Nhlh2 expression in midbrain and cerebellum from PWScr mice which lack Snord116 [39], and others have found no differences, including our own work with that mouse model.

The authors should explain in more details what type of stress(es) was (were) applied during Kummerfeld et al (39) study. Did the authors and others investigate the expression of Nhlh2 in the midbrain and cerebellum of Snord116del mice? Please provide relevant references. Did the authors refer to PMID: 35215509, which reported no differences in hypothalamic expression of Nhlh2 in wild type and Snord116del mice? If I am not mistaken Kummerfeld et al (39) also did not observe differences in hypothalamic expression.

All of the above point should be clarified in the review.

P10L321-324 High expression of SNHG14 message is also present in the lateral habenula (Fig. 5D)

What did the authors mean by these sentences? Did the authors detect human transcript in mouse brain? Did the authors detect Snurf/Snrpn part of Snhg14 transcript(s)? or maybe the observed hight expression gives us some message.

Please correct this part accordingly (including the Fig5 legend).

P10L334-332 First, the probe set used was developed commercially by ACD/Biotechne (Newark, CA) to detect full, unprocessed transcripts from the locus.

According to the information given in the legend of Figure5 (base pairs 2818-4022 of the NCBI reference sequence NR_146211.1), the probe was designed to detect the processed (spliced) transcript!

P10L337-338 Nucleolar-specific co-expression of Snhg14 with its possible targets may also be invaluable in clarifying functional information of when and where the ncRNAs are working.

Do the authors really mean that PWS-locus (or Snhg14) is specifically transcribed in nucleoli?....

I did spend a lot of time reviewing this manuscript and I hope that all the above comments will be helpful to the authors, but unfortunately, I no longer have the time or energy to rewrite this paper.

Just one quick remark on “Single nucleotide variant analysis in SNHG14 and its hosted transcripts

This chapter is extremely speculative and anyone familiar with the UCSC browser can easily get this information.

I would rather concentrate on reviewing what is known scientifically.
